# Teach2Eval: An Interaction-Driven LLMs Evaluation Method via Teaching Effectiveness

**Yuhang Zhou**[1,2][*] **Xutian Chen**[1][*] **Yixin Cao**[1,2][†] **Yuchen Ni**[1,2] **Yu He**[1,2] **Siyu Tian**[1]
**Xiang Liu**[3] **Yunwen Chen**[4] **Guangnan Ye**[1,2][†] **Xipeng Qiu**[1,2] **Hongfeng Chai**[1]

[1]Fudan University [2]Shanghai Innovation Institute [3]NYU Shanghai [4]DataGrand Inc.

## Abstract

Recent progress in large language models (LLMs) has outpaced the development of effective evaluation methods. Evaluating LLMs with static, task-specific benchmarks is increasingly fragile due to contamination and saturation, and it fails to capture interactive reasoning. We introduce Teach2Eval, which reframes evaluation as teaching: a candidate model guides weaker students, and the students' gains constitute the score. This interaction yields robustness to contamination and exposes orthogonal abilities with fine-grained metrics across Application, Judgment, Guidance, and Reflection. The framework scales automatically by exploiting natural error distributions from weak students, requiring neither bespoke rubrics nor human graders. Across 33 LLMs and 60 datasets, Teach2Eval achieves Spearman above 0.97 with human-preference leaderboards (e.g., Chatbot Arena/LiveBench), surpassing direct baselines, while offering actionable training signals (capability hierarchies, early overfitting) at low cost. We open-source our code and data at https://github.com/zhiqix/Teach2Eval.

## 1 Introduction

Large language models (LLMs) have advanced rapidly (OpenAI, 2023; Touvron et al., 2023), evolving from passive predictors to agentic systems capable of multi-step planning and interactive collaboration, yet evaluation has not kept pace. Current practice largely falls into two camps: (i) static and task-specific benchmarks, such as GSM8K, MATH, MMLU, and BIG-bench (Cobbe et al., 2021; Hendrycks et al., 2021; 2020; Srivastava et al., 2022), and (ii) bespoke agent environments such as code sandboxes(Dou et al., 2024; Zhou et al., 2023b), social simulations(Zhang et al., 2025). Both directly grade a model's problem-solving on specific tasks or environments, making results tightly coupled to test content and thus vulnerable to data contamination, saturation, and overfitting. This tension prompts a natural question: *Beyond endlessly refreshing datasets or engineering new test environments, can we adopt an evaluation paradigm that is less item-dependent and more suitable for the interactive, agentic nature of modern LLMs?*

We propose a shift in perspective: instead of asking *How well does a model solve tasks directly?*, we ask *How well can a model teach others to solve tasks?* Teaching-based paradigms have been widely applied to LLMs, some works distill knowledge from stronger models (Tian et al., 2024; He et al., 2024), while others use strong models to generate critics or rationales to augment data for weaker models Ying et al. (2024). Building on this line, we evaluate a model's capability by the performance gains it elicits in student models, using those gains as an indirect but informative signal of the teacher's competence.

Inspired by the Feynman Technique (Feynman, 2018), we introduce *Teach2Eval*, a general and automated evaluation framework that measures an LLM's multi-dimensional abilities through its effectiveness as a teacher, as shown in Figure 1. Instead of answering questions directly, the teacher provides feedback, corrections, and iterative guidance that help the student improve; the resulting student gains serve as the core metric. Crucially, this reframes evaluation as an interactive

---

[*]Equal contribution.
[†]Corresponding authors: yxcao@fudan.edu.cn; yegn@fudan.edu.cn

Figure 1: Comparing two evaluation methods. The left shows the use of static benchmarks to directly evaluate LLMs; On the right is Teach2Eval, which generates guidance to enhance the ability of the weak model as an indirect metric for LLMs evaluation.

process encompassing Judgment, Guidance, and Reflection abilities. By disentangling these abilities, Teach2Eval moves beyond task-specific correctness and captures fine-grained cognitive traits that are orthogonal to existing benchmarks.

This teaching-based evaluation method offers several advantages. **Robustness to contamination (Insight 1 in Section 5):** Unlike static tests that reward memorized answers (Wei et al., 2023; Zhou et al., 2023a), the teacher never observes choices and gold labels; instead, it must diagnose errors in a student's free-form reasoning and propose generalizable fixes, removing the common option-matching channel (Yang et al., 2023; Oren et al., 2023) and making memorization irrelevant. In practice, multiple weaker students with heterogeneous failure modes, shuffled distractors, and perturbed phrasings of the same item together create a moving target that is difficult to overfit. **Comprehensive abilities (Section 4.3):** Rather than relying on item design as in traditional benchmarks, Teach2Eval does not rely on the questions themselves (orthogonal to existing benchmarks). It grounds the evaluation in interaction, and the teaching process naturally involves several practical abilities. We categorize them into four-level fine-grained metrics based on Bloom's taxonomy (Krathwohl, 2002): Application, Judgment, Guidance, and Reflection. **Scalable automation (Insight 4 in Section 5):** Teach2Eval needs neither human graders nor hand-crafted suites, and runs at scale over large datasets and model pools. Weak students generate realistic errors, removing the annotation bottleneck. Moreover, this framework leverages weaker models to assess stronger ones by translating teaching effectiveness into the measurable signal of student improvement, which is a goal long regarded as hard to achieve (Khan et al., 2024).

Empirical results validate the effectiveness of this method. We measure how much student models improve under guidance across diverse datasets covering knowledge, reasoning, understanding, and multilingual domains, with a total of 60 datasets collected. Across 33 leading LLMs, Teach2Eval achieves Spearman correlations above 0.97 with dynamic human-preference benchmarks such as Chatbot Arena and LiveBench. In addition, we evaluate the convergence and robustness of this method based on a random combination of student models and longer interaction turns. We present some advantages of using this evaluation method to guide model training in Section 5. For instance, by analyzing the evolution of the two capabilities during model training, our method provides clear directions for preventing model overfitting. These findings not only demonstrate strong consistency with existing leaderboards in lower cost, but also highlight the framework's ability to surface nuanced differences in higher-order abilities.

We make the following contributions:

- We reframe LLM evaluation as a teaching problem, proposing Teach2Eval as an indirect, interaction-driven framework that measures models by their ability to improve weaker students.

- Our approach provides not only an effective overall ranking of models at low cost, but also a richer view of LLM capabilities beyond traditional benchmarks.

- Comprehensive empirical validation: Through experiments on 33 LLMs and 60 datasets, we show that Teach2Eval achieves correlations above 0.97 with human-based rankings, mitigates contamination risks, and reveals capability hierarchies.

## 2 RELATED WORK

### 2.1 LLM EVALUATION

Early LLM evaluation was task-centric (Cao et al., 2025), scoring models on fixed-format benchmarks (GLUE (Wang et al., 2018), HellaSwag (Zellers et al., 2019), PIQA (Bisk et al., 2020)) for tasks like classification, extraction, inference; reproducible and metric-clear, but vulnerable to contamination, saturation, and weak coverage of open-ended tasks (Wei et al., 2023; Zhou et al., 2023a). Crowd-sourced evaluation emphasizes open settings and interaction diversity, e.g., DynaBench's adversarial collection (Kiela et al., 2021) and Chatbot Arena's crowd rankings (Chiang et al., 2024), mitigating leaderboard gaming and memorization; "LLM-as-a-Judge" further cuts cost and scales (Zheng et al., 2023). Teach2Eval differs in the source of supervision: rather than grading answers or relying on a judge, it measures student gains under teacher guidance as an *indirect* signal of teacher competence. Two design choices reduce contamination and overfitting: (i) teachers never see answer options or gold labels; (ii) multiple heterogeneous students, perturbed phrasings, and shuffled distractors create moving targets that weaken option-matching channels. The protocol is fully automated and cost-efficient, and its interaction-centered signal aligns with real LLM usage.

### 2.2 TEACHING PARADIGM

The teacher–student paradigm originated with knowledge distillation, transferring a large teacher's soft targets to a smaller student to balance accuracy and efficiency (Gou et al., 2021; Hinton et al., 2015). In LLMs, variants include multi-teacher (Tian et al., 2024), uncertainty-guided (He et al., 2024), and retrieval-augmented distillation (Zhang et al., 2023), as well as weak-to-strong transfer (Burns et al., 2023). Beyond distillation, simulated teaching leverages multi-agent classroom settings (Zhang et al., 2024; Shi et al., 2025), Dean–Teacher–Student pipelines (Liu et al., 2024), and student knowledge graphs for generating responses at different cognitive levels (Wu et al., 2025). Further studies (Ning et al., 2024) show that teaching itself can refine a model's reasoning and knowledge structuring. In con-

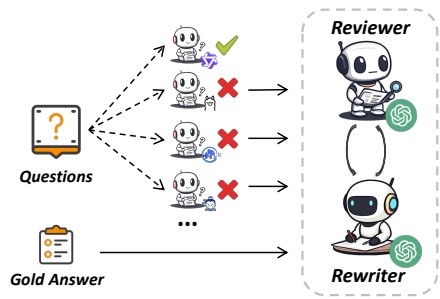

Figure 2: Automated MCQ conversion: use weaker models to simulate diverse, misleading options; employ GPT-4o as a rewriter and reviewer, convert questions into MCQ format.

trast to these role-playing paradigms, Teach2Eval allows the teacher model to employ any effective instructional strategy at its discretion and converts the teaching process into an evaluation signal, unifying capability orientation with interaction-centered evaluation.

## 3 METHOD

### 3.1 DATA CONSTRUCTION

To ensure the effectiveness and diversity of the evaluation, we collect 60 datasets, and categorize them into Knowledge, Reasoning, Understanding, and Multilingual domains, with detailed information provided in Appendix B. To obtain accurate, unbiased, and reproducible estimates on these tasks, we standardize all items into multiple-choice query (MCQ) form: MCQs enable consistent, scalable scoring without costly human judgment, while preserving difficulty by using hard distractors derived from real model mistakes (Gu et al., 2024; Li et al., 2024). Concretely, for each question we collect distractors from weak models, combine them with the gold answer, and employ GPT-4o to normalize formatting and consistency; the correct option's position is randomized to avoid position bias (Pezeshkpour & Hruschka, 2023), as shown in Figure 2). Student models are scored only on the finalized MCQs, while the teacher model provides guidance without ever seeing the answer choices, which preserves item difficulty and prevents option leakage. Finally, we stratify all items into five difficulty bands using the observed accuracies of Qwen-family models (Yang et al., 2024) of varying

sizes. These difficulty strata then support analysis LLMs' abilities across various difficulty levels, full statistics are reported in Appendix B.

We introduce Teach2Eval, an indirect evaluation protocol that judges an LLM by how much it lifts weaker students under blinded-choice tutoring. Rather than grading the teacher's own answers, we quantify the performance gain it elicits in student models across diverse tasks, as shown in Figure 3.

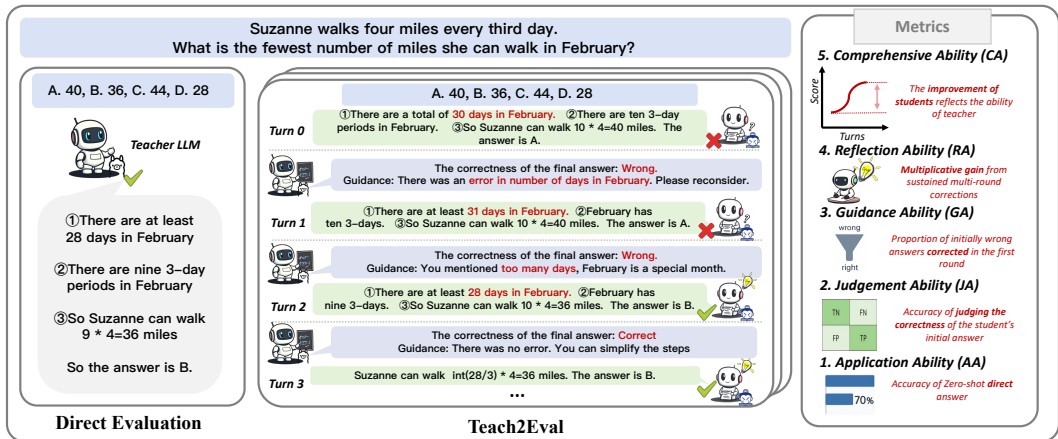

Figure 3: Overview of Teach2Eval. It requires that the LLM guides students to answer questions, enabling a assessment of LLM's fine-grained abilities that are orthogonal to current benchmarks.

## 3.2 TEACH2EVAL

For each data item $d_n$ with gold answer $y_n$ from dataset $\mathcal{D}$, student $\mathcal{S}_m$ first answers using only the question and the options, producing $a_{m,n,0}$. The teacher $\mathcal{T}$ then observes the question and the full transcript without the answer choices, and returns a judgment $j_{m,n,t}$ and guidance $g_{m,n,t}$. The student updates its answer accordingly:

$$a_{m,n,t} = \mathcal{S}_m\big(d_n, a_{m,n,t-1}, g_{m,n,t}\big), \quad t \geq 1, \qquad a_{m,n,0} = \mathcal{S}_m(d_n).$$

The teacher's output at round $t$ is

$$j_{m,n,t}, g_{m,n,t} = \mathcal{T}\big(d_n, H_{m,n,t-1}\big),$$

where $H_{m,n,t-1} = \{a_{m,n,0}, j_{m,n,1}, g_{m,n,1}, \ldots, a_{m,n,t-1}\}$. This blinded setup preserves open-ended reasoning while preventing option-level leakage. Let $\mathbb{I}[\cdot]$ be the indicator function. The round-$t$ accuracy gain for student $\mathcal{S}_m$ is

$$\Delta P_t(\mathcal{S}_m) = \frac{1}{|D|} \sum_{n=1}^{|D|} \Big( \mathbb{I}[a_{m,n,t} = y_n] - \mathbb{I}[a_{m,n,t-1} = y_n] \Big).$$

Given a budget of $T$ tutoring rounds, we report the mean cumulative lift across $M$ students as **Comprehensive Ability (CA)** metric:

$$\text{CA} = \frac{1}{M} \sum_{m=1}^{M} \sum_{t=1}^{T} \Delta P_t(\mathcal{S}_m).$$

## 3.3 ABILITY TAXONOMY AND METRICS

Grounded in Bloom's taxonomy (Krathwohl, 2002) and our blinded-choice tutoring protocol, we factorize the teacher's capability into four components: Application, Judgment, Guidance, and

Reflection, which together explain the observed lift in Comprehensive Ability (CA), we demonstrate this in Appendix A. Formally:

**Application Ability (AA).** The teacher's ability to answer the questions directly. This is the basic ability of the LLM to use its knowledge to provide answers. We denote it as the accuracy of the LLM when performing Zero-Shot testing:

$$
\text{AA} \;=\; \frac{1}{|\mathcal{D}|} \sum_{n=1}^{|\mathcal{D}|} \mathbb{I}\big[\, \mathcal{T}(d_n) = y_n \,\big].
$$

**Judgment Ability (JA).** The teacher's ability to correctly judge a student's answer given the question and transcript (but not the options). We evaluate judgments on the first round:

$$
\text{JA} \;=\; \frac{1}{M} \sum_{m=1}^{M} \frac{1}{|\mathcal{D}|} \sum_{n=1}^{|\mathcal{D}|} \mathbb{I}\Big[\, J_{m,n,1} = \mathbb{I}[\, a_{m,n,0} = y_n\,] \,\Big],
$$

where $J_{m,n,1}$ is the teacher's judgment in the first round, and $\mathbb{I}[a_{m,n,0} = y_n]$ is the ground-truth correctness of the initial answer $a_{m,n,0}$ for item $d_n$.

**Guidance Ability (GA).** Effectiveness of the first round guidance at repairing initially incorrect answers. Let $\mathcal{D}_m^{\text{inc}} = \{\, n :\; a_{m,n,0} \neq y_n \,\}$. Then

$$
\text{GA} \;=\; \frac{1}{M} \sum_{m=1}^{M} \frac{\sum_{n \in \mathcal{D}_m^{\text{inc}}} \mathbb{I}\big[\, a_{m,n,1} = y_n \,\big]}{|\mathcal{D}_m^{\text{inc}}|}.
$$

**Reflection Ability (RA).** Beyond first-round guidance, RA measures multi-round refinement as a *multiplicative* repair rate. For student $\mathcal{S}_m$ and round $t \geq 2$, let $C_{m,t-1}$ is the number of items currently correct before round $t$; $\text{Fix}_{m,t}$ is the number of previously incorrect items repaired to correct at round $t$; $\text{Reg}_{m,t}$ is the number of previously correct items that regress to incorrect at round $t$.So he per-round reflection multiplier is

$$
r_{m,t} = \frac{C_{m,t-1} + \text{Fix}_{m,t} - \text{Reg}_{m,t}}{C_{m,t-1}} = 1 + \frac{\text{Fix}_{m,t} - \text{Reg}_{m,t}}{C_{m,t-1}}, \quad t \geq 2,
$$

with $r_{m,t} = 1$ when $C_{m,t-1} = 0$ to avoid division by zero. We aggregate across later rounds by multiplication:

$$
\text{RA}_m = \prod_{t=2}^{T} r_{m,t} - 1, \qquad \text{RA} = \frac{1}{M} \sum_{m=1}^{M} \text{RA}_m.
$$

Table 1: Correlation of Teach2Eval with Chatbot Arena and LiveBench.

| Evaluation Method | Chatbot Arena | | Livebench | |
|---|---|---|---|---|
| | Spearman Cor | Kendall's Tau | Spearman Cor | Kendall's Tau |
| Direct Evaluation | 0.734 | 0.558 | 0.861 | 0.695 |
| Teach2Eval | **0.944** | **0.853** | **0.975** | **0.886** |

## 4 EXPERIMENTS

### 4.1 EXPERIMENT SETUP

In order to expand the evaluation scope of our method as much as possible, we define the selection criteria for weak student models as follows: strong instruction-following abilities but weak original application abilities. With the rapid progress of on-device LLMs, most mainstream on-device models satisfy this criterion and thus serve as our default student pool. Therefore, we select four student models: LLaMA3.2-1B (Dubey et al., 2024), Qwen2.5-1.5B (Yang et al., 2024), MiniCPM-2B (Hu et al., 2024), and InternLM2.5-1.8B (Cai et al., 2024). Additionally, we select 30 leading models, including various families for evaluation. Detailed information can be found in Appendix C. The experiments are conducted using the VLLM 0.6.4 framework for inference, with the following settings: temperature = 0.0, max_tokens = 8k, on four H100 GPUs.

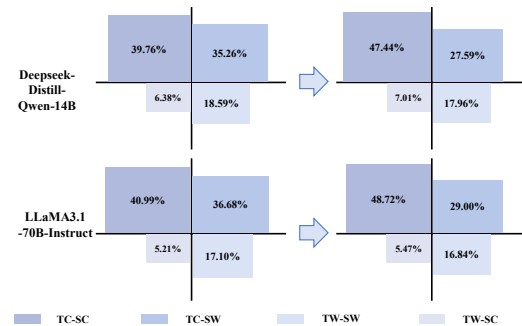

Figure 4: The confusion matrix comparison between DeepSeek and LLaMA models shows the left side without guidance and the right side after 3 turns of guidance. T and S represent teacher and student models, while C and W represent correct and wrong models, respectively.

Table 2: The performance of the LLMs under Teach2eval.

| Model | Overall | | | | | Knowledge | Reasoning | Understanding | Multilingual |
|---|---|---|---|---|---|---|---|---|---|
| | Comprehensive Ability | Application Ability | Judgment Ability | Guidance Ability | Reflection Ability | | | | |
| Claude-Sonnet-4-thinking-20250514 | 14.12 | 85.47 | 76.22 | 23.40 | 8.35 | 14.69 | 15.50 | 9.46 | 21.12 |
| o4-mini-2025-04-16 | 12.38 | 85.60 | 79.23 | 19.73 | 6.86 | 14.38 | 13.98 | 6.73 | 18.02 |
| Gemini-2.5-pro | 11.77 | 85.40 | 77.00 | 17.26 | 9.20 | 14.07 | 10.14 | 9.29 | 21.90 |
| Qwen3-30B-A3B | 11.63 | 76.92 | 65.64 | 26.19 | 5.05 | 13.26 | 12.53 | 6.69 | 18.54 |
| Qwen3-32B | 10.35 | 74.17 | 73.89 | 21.08 | 3.62 | 8.61 | 13.30 | 5.42 | 16.36 |
| Qwen2.5-72B-Instruct | 10.07 | 78.91 | 74.49 | 20.89 | 3.74 | 11.72 | 11.26 | 5.53 | 14.61 |
| DeepSeek-V3 | 9.84 | 75.20 | 73.78 | 19.45 | 5.05 | 10.29 | 11.12 | 5.71 | 15.07 |
| DeepSeek-R1-Distill-Llama-70B | 9.54 | 76.46 | 73.79 | 18.80 | 3.51 | 10.74 | 11.11 | 5.76 | 11.38 |
| DeepSeek-R1-Distill-Qwen-32B | 8.68 | 77.56 | 75.00 | 16.67 | 3.62 | 9.12 | 10.04 | 5.51 | 11.25 |
| GPT-4o-0806 | 8.55 | 76.97 | 75.70 | 15.60 | 6.86 | 9.73 | 9.96 | 4.71 | 10.77 |
| DeepSeek-R1-Distill-Qwen-14B | 8.31 | 75.03 | 72.52 | 16.71 | 2.91 | 7.86 | 10.32 | 5.12 | 10.20 |
| Llama-3.3-70B-Instruct | 7.97 | 77.69 | 75.05 | 13.27 | 5.74 | 8.61 | 9.44 | 5.19 | 8.53 |
| Phi-4 | 7.69 | 76.58 | 73.20 | 15.53 | 3.14 | 7.80 | 9.50 | 4.42 | 9.49 |
| Qwen3-8B | 7.64 | 63.66 | 74.24 | 18.20 | 1.24 | 6.73 | 9.28 | 5.04 | 10.39 |
| Qwen2.5-Coder-32B-Instruct | 7.40 | 76.20 | 72.33 | 16.22 | 3.07 | 7.02 | 8.66 | 5.13 | 9.64 |
| GPT-4o-mini | 7.20 | 72.20 | 72.17 | 14.62 | 3.68 | 7.62 | 8.91 | 4.94 | 5.23 |
| Llama-3.1-70B-Instruct | 7.13 | 75.10 | 71.73 | 15.31 | 3.74 | 8.22 | 7.84 | 4.62 | 9.04 |
| Qwen2.5-32B-Instruct | 6.85 | 79.23 | 74.09 | 14.33 | 3.74 | 7.82 | 7.30 | 4.80 | 8.82 |
| Qwen2-72B-Instruct | 6.16 | 74.20 | 72.45 | 13.83 | 3.43 | 7.11 | 6.35 | 4.89 | 6.98 |
| DeepSeek-R1-Distill-Qwen-7B | 5.61 | 61.93 | 53.82 | 12.53 | 1.67 | 3.48 | 8.68 | 2.62 | 6.31 |
| Qwen2.5-14B-Instruct | 5.49 | 76.38 | 71.13 | 12.88 | 2.73 | 6.60 | 5.76 | 4.03 | 6.11 |
| Llama-3-70B-Instruct | 4.17 | 73.03 | 70.89 | 10.36 | 2.30 | 5.09 | 3.78 | 4.02 | 4.14 |
| Qwen2.5-7B-Instruct | 3.93 | 72.26 | 66.95 | 9.93 | 1.96 | 4.08 | 4.38 | 2.85 | 4.88 |
| Yi1.5-34B-Chat | 3.93 | 68.61 | 61.74 | 8.93 | 2.17 | 3.31 | 4.44 | 3.46 | 4.52 |
| DeepSeek-R1-Distill-Llama-8B | 3.60 | 63.87 | 60.45 | 11.15 | 0.41 | 2.34 | 5.59 | 1.90 | 3.11 |
| Yi1.5-9B-Chat | 2.56 | 64.19 | 63.55 | 12.05 | 0.87 | 2.66 | 2.82 | 2.06 | 2.68 |
| InternLM2.5-20B | 2.16 | 64.94 | 62.20 | 8.12 | -0.03 | 2.14 | 2.25 | 2.19 | 1.67 |
| Gemma-2-27b-it | 1.89 | 70.38 | 67.63 | 8.05 | 0.58 | 1.74 | 1.82 | 1.75 | 2.95 |
| Llama-3.1-8B-Instruct | 1.79 | 62.37 | 54.12 | 10.13 | 0.55 | 2.37 | 1.55 | 1.46 | 2.42 |
| Gemma-2-9b-it | 1.70 | 65.63 | 64.77 | 7.79 | 0.40 | 1.43 | 1.57 | 1.46 | 3.52 |
| Llama-3-8B-Instruct | 1.37 | 59.66 | 53.79 | 9.69 | 0.42 | 1.90 | 1.14 | 1.31 | 1.37 |
| InternLM2.5-7B | 1.01 | 55.73 | 52.77 | 5.06 | -1.02 | 0.77 | 0.97 | 1.24 | 1.02 |
| Yi1.5-6B-Chat | 0.91 | 57.12 | 54.08 | 9.34 | -0.85 | 1.67 | 0.82 | 0.28 | 1.40 |

## 4.2 MAIN RESULT

We compare 33 models based on the experimental setup described above, and the detailed results are provided in Appendix E. To verify the effectiveness of our evaluation, we compare our results with two prominent leaderboards. To ensure the consistency of the data distribution, we compare our reasoning scores with the Chatbot Arena Math benchmark and our overall scores with the Livebench Reasoning, Math, and Language benchmark. The correlation coefficients are shown in Table 1. Compared with direct evaluation, our evaluation method achieves a higher correlation between two leaderboards, both above 0.94, and is also more cost-effective than two leaderboards.

As shown in Table 2, we find that Claude-4-Sonnet, o4-mini and Gemini-2.5-pro perform the best across all four types of tasks, significantly outperforming the other models. The next best models are Qwen2.5-72B-Instruct and the DeepSeek series, which show similar performance. DeepSeek-R1-Distill-Qwen-14B and Phi-4 have only 14B parameters, demonstrating that even models of this

scale can achieve exceptional performance. For models with 8B parameters or smaller, DeepSeek-R1-Distill-Qwen-7B shows strong performance, especially in the reasoning tasks, where its ability even rivals that of the 70B variant from the same family. Due to DeepSeek-R1-Distill-Qwen-14B's impressive performance with models below 20B, we compare its confusion matrix with that of Llama3.1-70B-Instruct, as shown in Figure 4. The results indicate that DeepSeek-R1-Distill-Qwen-14B can teach student models more effectively within the capacity of teacher models, showcasing its higher-level abilities.

We further examine whether our evaluation remains effective across the full capability spectrum and whether it saturates at the high end. Figure 5 visualizes the relationship between our metric and two human preference leaderboards without any binning. Teach2Eval exhibits a tight, monotonic alignment with both LiveBench and Chatbot Arena, with substantially smaller dispersion than Direct Evaluation. In contrast, Direct Evaluation shows noticeable score compression and larger residuals in the high-capability region. These indicate that Teach2Eval remains discriminative across the full range and does not exhibit saturation at the top end until now.

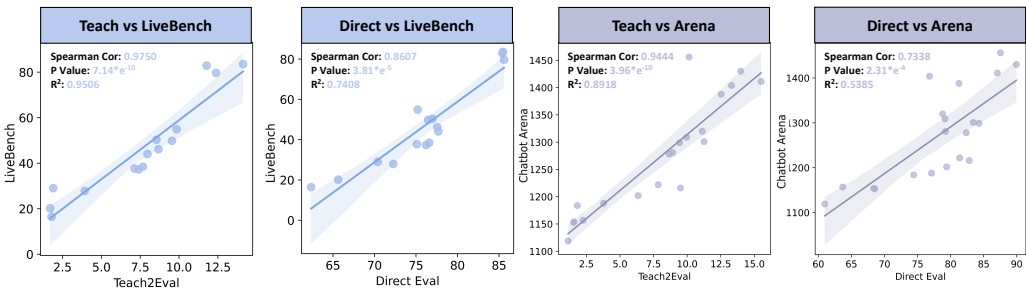

Figure 5: Alignment with human preference leaderboards. Four panels show scatter plots with linear fits comparing Teach2Eval and Direct Evaluation against LiveBench and Chatbot Arena. Teach2Eval exhibits stronger monotonic alignment and smaller residuals on both leaderboards.

## 4.3 ABILITY DIMENSIONAL ANALYSIS

To further explore the factors influencing the overall performance of LLMs, we use the classification method designed in Section 3.3 to assess the four capabilities of all LLMs and compare them with their Comprehensive Ability. Table 2 presents a comparison of the four capabilities with the Comprehensive Ability across all datasets. As shown in Figure 6, we observe that the correlation between the Comprehensive Ability and the four capabilities gradually increases. The Application Ability obtained through traditional direct evaluation has a relatively low correlation with our Comprehensive Ability, while the correlation for the higher-order abilities is above 0.9. For some models that show anomalies in direct evaluations, such as Qwen2.5-32B-Instruct, which achieves the especially high Application Ability, our evaluation method reveals its true capabilities.

All models exceed 50% in judgment ability, indicating that all LLMs possess judgment capabilities for student models. However, there are significant differences among the models in the comparisons for Guidance and Reflection abilities. For Reflection Ability, the models with the worst performance, Yi-1.5-6B-Chat and InternLM2.5-7B, exhibit inconsistent reflection, performing worse than the other models.

## 4.4 ABLATION EXPERIMENT

To verify the robustness and convergence of Teach2Eval, we conduct three ablation experiments.

**Ablation 1: No bias in random selection of weak student models.** To assess potential bias from student model selection, we evaluate the stability of results when one of the four student models is randomly removed. This yields four combinations of three models each. The resulting correlations with Chatbot Arena (0.926–0.936) and LiveBench (all above 0.94) confirm the robustness and impartiality of the evaluation mechanism.

**Ablation 2: Higher correlation and scale-robustness vs. direct evaluation.**

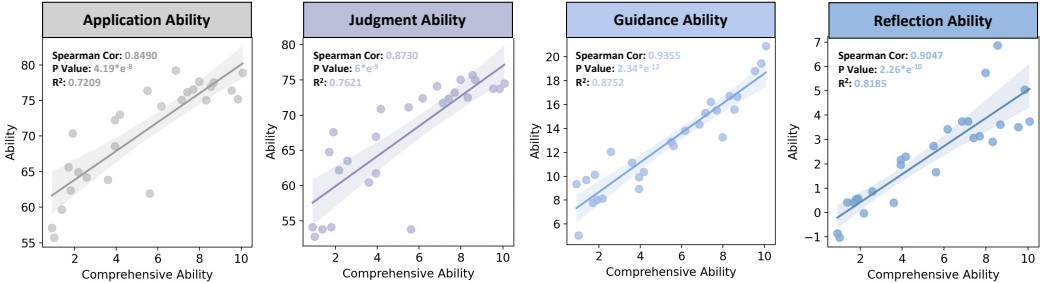

Figure 6: The correlation coefficient between Comprehensive Ability and the four abilities.

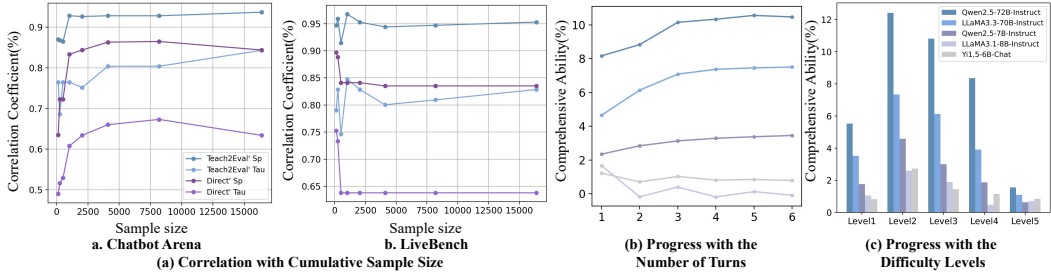

Figure 7: (a) Correlation on two leaderboards vs. sample size for Teach2Eval and direct evaluation.(b) Student performance gain vs. number of guidance rounds.(c) Progress across difficulty tiers by different teacher models.

To verify faster and earlier convergence, we compare Teach2Eval with direct evaluation under cumulative sampling (each point augments the previous draw), with results shown in Figure 7(a). On Chatbot Arena, Teach2Eval reaches high and stable correlations quickly: Spearman > 0.92, Kendall's Tau > 0.8; while direct evaluation lags and is less stable with mild degradation. LiveBench shows the same trend: Teach2Eval sustains 0.95 Spearman and 0.83 Tau with smooth, monotonic gains, whereas direct evaluation's Tau drops early and plateaus near 0.64. Across sample sizes, Teach2Eval consistently attains higher correlations and earlier convergence.

**Ablation 3: student model improvement tends to stabilize after multiple guidance turns.** To evaluate the convergence and Reflective Ability of LLMs, we select five representative models, each providing six rounds of guidance to student models (performance shown in Figure 7(b)). While high-performing models show initial gains, the improvement generally plateaus as the number of rounds increases. Lower-performing models like LLaMA3.1-8B and Yi1.5-6B reach a plateau even earlier, indicating limited reflective capability. These results confirm the convergence behavior of our method. In our main experiments, we adopt the performance after three turns as the final metric, as most models have nearly converged by then.

**Ablation 4: student model improvement tends to stabilize after multiple guidance turns.** To evaluate the convergence and Reflective Ability of LLMs, we select five representative models, each providing six rounds of guidance to student models (performance shown in Figure 7(b)). While high-performing models show initial gains, the improvement generally plateaus as the number of rounds increases. Lower-performing models like LLaMA3.1-8B and Yi1.5-6B reach a plateau even earlier, indicating limited reflective capability. These results confirm the convergence behavior of our method. In our main experiments, we adopt the performance after three turns as the final metric, as most models have nearly converged by then.

**Ablation 5: Teach2Eval is robust to teacher "answer revelation".** In Teach2Eval, teacher's prompt does not restrict teaching style or whether the final answer may be stated. To test whether answer revelation affects conclusions, we trained an answer detector (based on Qwen3-32B) to label each interaction as Answer Revealed or No Answer Revealed, then computed the ranking of student gain within each subset, as shown in Table 3 Correlating these rankings with two baselines shows that, regardless of revelation, the relevance with Teach2Eval is consistently higher than with

Direct Evaluation. This indicates Teach2Eval rewards diagnosis–guidance–reflection rather than mere answer disclosure and is robust to prompt strategies that "just tell the answer."

Table 3: Correlation of rankings (with/without answer revelation) with Teach2Eval and Direct Eval.

| Subset | vs. Teach2Eval | | vs. Direct Evaluation | |
|---|---|---|---|---|
| | Spearman Cor | Kendall's Tau | Spearman Cor | Kendall's Tau |
| No Answer Revealed | **0.947** | **0.805** | 0.725 | 0.564 |
| Answer Revealed | **0.921** | **0.791** | 0.815 | 0.633 |

## 5 INSIGHTS

In addition to providing effective model rankings, Teach2Eval can also offer fine-grained indicators, which can transform static task-specific evaluations into dynamic capability-based evaluations. Such analysis of the model can naturally guide further training and refinement.

**Insight 1: Teach2Eval reduces evaluation sensitivity to data contamination.** To probe contamination effects, we construct a distillation-evaluation subset as follows: we prompt GPT-4o to answer the question pool, keeping items matching gold labels, and uniformly sampling 3,000 for distillation. Six LLMs fine-tuned on this subset are evaluated in a two-dimensional space: Application Ability (AA) and Comprehensive Ability (CA). As in Figure 8, most distilled LLMs show higher AA but lower CA, while stronger baselines (e.g., Qwen2.5-32B, Phi-4) change little in CA. This pattern suggests Teach2Eval is less susceptible than direct answer scoring to inflation arising from contaminated supervision, and that tracking training trajectories in this metric space can provide early feedback to curb overfitting or gains caused by contamination.

**Insight 2: The Scaling Law remains valid for higher-order capabilities.** We conduct a study on the Scaling Law (Kaplan et al., 2020) of five model families, including Qwen, Deepseek, LLaMA, InternLM, and Yi, as shown in Figure 10. Our findings show that, across the overall dataset, the Comprehensive Ability increases with model size within each family. However, for the DeepSeek-Distill family models, variations in the size of the base models (Qwen or Llama) lead to fluctuations in their higher-order capabilities, which is more consistent with consensus. In terms of Application Ability, these models follow the trend that larger sizes yield better performance, which may result in incorrect evaluations during training, potentially leading to models performing poorly in reality. This suggests that while models may appear to follow the Scaling Law in Application Ability, their true capabilities can differ, we can use higher-order capabilities to assess models' genuine performance.

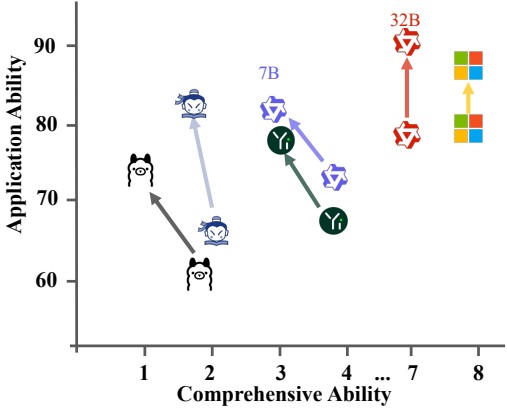

Figure 8: Position changes of six LLMs on the AA–CA axes before and after fine-tuning (arrows from base to fine-tuned).

**Insight 3: Reasoning enhancement is effective in many tasks, but its impact varies across different tasks.** Currently, reasoning LLMs like OpenAI-o1 (Jaech et al., 2024) and Deepseek-R1 (Guo et al., 2025) are gaining popularity. These models use test-time scaling to further enhance their reasoning abilities, which has attracted significant attention. To assess the abilities of these reasoning models, we also select the Deepseek family, which were distilled based on Qwen and LLaMA models. In Figure 9, we compare the reasoning LLMs with their base models. Our findings show that the reasoning models outperform the base models across all tasks, where the 14B reasoning model surpasses the 32B base model, especially in the Reasoning and Multilingual tasks. The

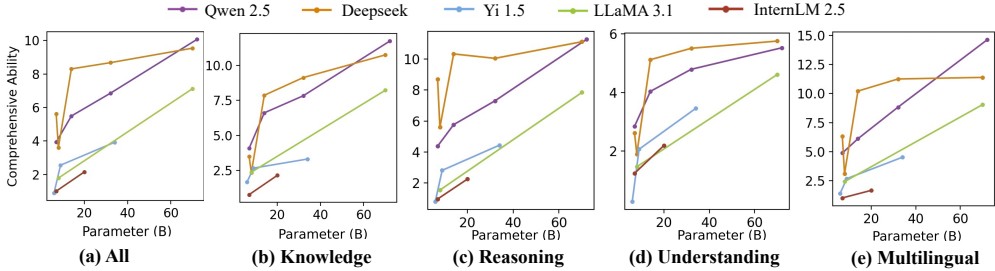

Figure 10: The trend of Comprehensive Ability changing with the parameters for model-families.

improvement in the Knowledge task is less pronounced, and only minimal gains are observed in the Understanding task.

**Insight 4: The low-order capabilities of weaker models can be leveraged to inversely evaluate the high-order capabilities of stronger models.** As models become increasingly capable, it becomes challenging to find stronger models or sufficiently broad datasets for effective evaluation. Previous research (Khan et al., 2024) has explored whether the critical capabilities of weaker models can be used to judge the responses of stronger models; however, such subjective evaluations are inherently limited by the capacity ceiling of the weaker models. Teach2Eval leverages the response-generation capability of weak models, elevating the evaluation ceiling to match that of the stronger teacher models. This enables more accurate and effective assessment of model capabilities, making it particularly suitable for potential future AGI scenarios.

**Insight 5: Medium-difficulty problems are more successfully guided to improvements.** To analyze the relationship between guidance and model capability across data difficulty levels, we calculate the guidance improvement ratios for each difficulty set. The statistical details can be found in Appendix B, and we also present five models in Figure 7(c). We find that the guidance effect is most effective on data of medium difficulty, and this effect exhibits a decreasing trend as the difficulty level increases. Strong models show robust guidance abilities across all difficulty levels. This indicates that further attention needs to be paid to the model's understanding of simple problems during model training, which may be overlooked in favor of more complex cases.

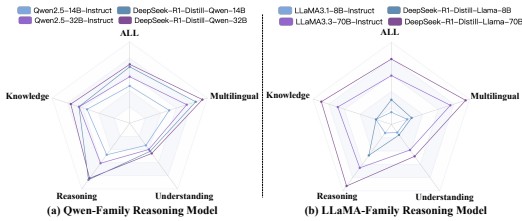

Figure 9: The Comprehensive Ability of reasoning LLMs and their base models varies across different types of tasks, and reasoning LLMs have shown improvements in various types, especially in reasoning tasks.

## 6 CONCLUSION

We presented Teach2Eval, an interaction-driven evaluation protocol that scores an LLM by how much it improves weaker students under blinded guidance. This paradigm mitigates contamination, scales without human graders, and exposes higher-order abilities: Judgment, Guidance, and Reflection, beyond direct answer accuracy, yielding strong alignment with human-preference leaderboards at lower cost. Empirically, Teach2Eval converges quickly, surfaces nuanced capability differences obscured by static tests, and provides actionable signals for training and model selection. Future work includes expanding model pools and task coverage, linking metrics to downstream agent performance, and exploiting Teach2Eval signals to train models.

## 7 THE USAGE OF LLM

We used large language models (LLMs) only for light copy-editing of grammar and style in a few places. No technical content, analyses, or results were generated by LLMs; all ideas and writing were authored and verified by the authors.

## 8 ETHICS STATEMENT

This work complies with the ICLR Code of Ethics. No human-subjects research or animal experimentation was conducted. All datasets were used under their licenses and usage guidelines; no personally identifiable or sensitive data were collected or processed. We took steps to reduce potential biases and to avoid discriminatory outcomes. The study poses no known privacy or security risks. We will release code and documentation to support transparency and reproducibility.

## 9 REPRODUCIBILITY STATEMENT

We describe the parameters and software versions of our experiments in Section 4.1 of this article and record the prompt in Appendix D. We document the full training and evaluation pipeline, including hyperparameters, checkpoints, and ablation configurations, and we enable deterministic flags where supported. All datasets used in this paper are publicly available and referenced in the paper. We also describe the data construction method in Appendix B. Based on the information we have disclosed so far, we promise that all experiments can be reproduced.

## 10 ACKNOWLEDGEMENTS

This project was supported by the National Natural Science Foundation of China (NSFC) under Grant Nos. 72595845, 62576102, and 72595840.

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

## A  A DECONSTRUCTION OF COMPREHENSIVE ABILITY

We model the Comprehensive Ability (CA) of a teacher LLM $\mathcal{T}$ by the performance lift it imparts to a student model. We assume all metrics are averaged over the same student population, and for simplicity, we denote the student as $\mathcal{S}$. This comprehensive lift is influenced by three primary capabilities defined in Section 3.3: Judgment Ability (JA), Guidance Ability (GA), and Reflection Ability (RA). By deconstructing the performance gain, we can model the relationships between these abilities.

First, let $P_0(\mathcal{S}) = \frac{1}{|\mathcal{D}|} \sum_{n=1}^{|\mathcal{D}|} \mathbb{I}[a_{n,0} = y_n]$ be the student's initial accuracy. The accuracy gain after the first round of tutoring, $\Delta P_1(\mathcal{S})$, can be expressed as:

$$\Delta P_1(\mathcal{S}) \approx C_{\text{norm}} \cdot \text{GA} \cdot \Big(\text{JA} \cdot (1 - P_0(\mathcal{S})) + (1 - \text{JA}) \cdot P_0(\mathcal{S})\Big) - \alpha \cdot P_0(\mathcal{S}) \cdot (1 - \text{JA})$$

Here, $(1 - P_0(\mathcal{S}))$ is the student's initial error rate, and $(1 - \text{JA})$ is the teacher's judgment error rate. The term $\alpha$ is a regression factor accounting for instances where correct student answers are altered to be incorrect due to flawed guidance. The coefficient $C_{\text{norm}}$ is a normalization factor defined as the ratio of the student's initial error rate to the teacher's judgment error rate:

$$C_{\text{norm}} = \frac{1 - P_0(\mathcal{S})}{1 - \text{JA}}$$

Next, we consider the cumulative improvement over multiple tutoring rounds. The total lift across $T$ rounds is the Comprehensive Ability $\text{CA}(\mathcal{S})$. We propose a model where the teacher's Reflection Ability (RA) multiplicatively enhances the initial gain from the first round. This relationship can be expressed as a heuristic:

$$\text{CA}(\mathcal{S}) = \Delta P_1(\mathcal{S})\,(1 + \text{RA})$$

Through these equations, we posit that the teacher's Comprehensive Ability emerges from the interplay of its judgment, guidance, and reflection. Specifically, JA and GA interact to produce an initial performance lift, which is then amplified over subsequent rounds by the teacher's RA.

## B  DATASETS AND DATA CONSTRUCTION

To ensure the comprehensiveness and effectiveness of our evaluation, we collect 60 datasets and sample 15,000 pieces of data, classify them into four tasks: Knowledge, Reasoning, Understanding, and Multilingual. Figure 11 is the data statistical chart, and Table 4 is the data statistical table.

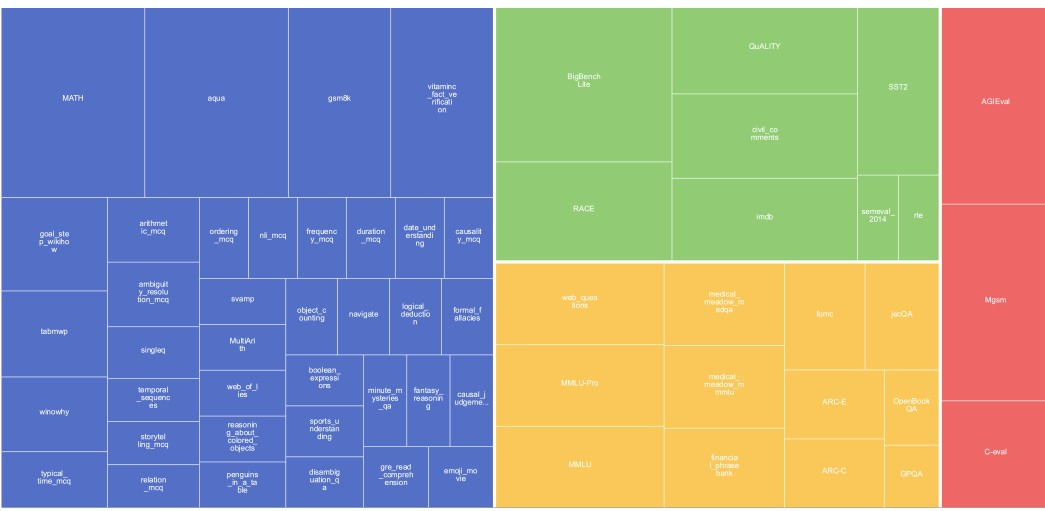

Figure 11: Dataset summary visualization, where blue represents Reasoning task, green represents Understanding task, yellow represents Knowledge task, red represents Multilingual task, and block size represents the number of samples in the dataset.

In order to modify all datasets to MCQ format, we use 10 weak models such as Qwen2.5-1.5B, Llama3.2-1B, etc., set the Temperature to 0.7, and randomly answer each question until we collected 3 incorrect answers. We use GPT-4o as the rewriter and reviewer, with gold answer as the correct answer and weak models answer as the incorrect answer for format conversion. We randomly place the correct answer positions during construction.

Afterwards, in order to classify all the data into difficulty categories, we used Qwen family models to answer each question sequentially from 1B to 14B. Classify the data into the difficulty category corresponding to the first correctly answered model, and set the question that all models cannot answer as the highest difficulty. Finally, divide it into five difficulty types, as shown in the Figure 12.

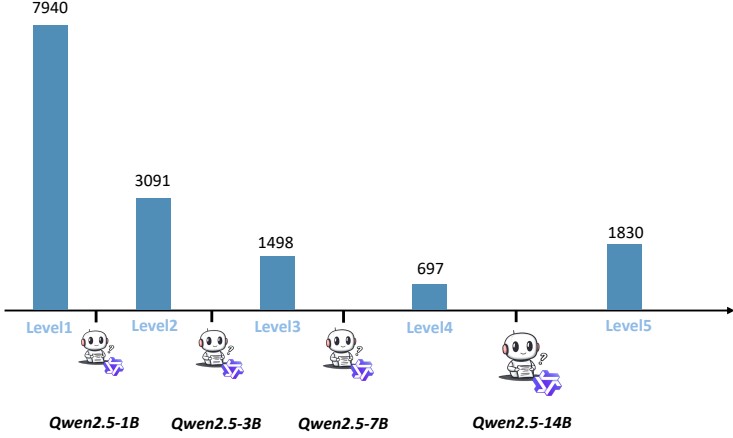

Figure 12: Use Qwen-family models for data difficulty classification.

Table 4: Dataset statistical information

| Task | Name | Num. | Name | Num. |
|---|---|---|---|---|
| Knowledge | ARC-C | 202 | ARC-E | 231 |
| | GPQA | 103 | MMLU | 433 |
| | MMLU-Pro | 433 | OpenBookQA | 115 |
| | web_questions | 433 | fomc | 246 |
| | financial_phrasebank | 289 | jecQA | 231 |
| | medical_meadow_mmmlu | 289 | medical_meadow_medqa | 289 |
| Multilingual | AGIEval | 578 | C-eval | 311 |
| | Mgsm | 577 | | |
| Understanding | BigBenchLite | 1156 | civil_comments | 462 |
| | imdb | 462 | QuALITY | 482 |
| | RACE | 1141 | rte | 64 |
| | semeval_2014 | 64 | SST2 | 421 |
| Reasoning | emoji_movie | 50 | gre_read_comprehension | 32 |
| | vitaminc_fact_verification | 879 | causal_judgement | 50 |
| | fantasy_reasoning | 50 | goal_step_wikihow | 393 |
| | minute_mysteries_qa | 50 | winowhy | 393 |
| | disambiguation_qa | 57 | sports_understanding | 57 |
| | boolean_expressions | 57 | formal_fallacies | 57 |
| | logical_deduction | 57 | navigate | 57 |
| | object_counting | 57 | penguins_in_a_table | 50 |
| | reasoning_about_colored_objects | 57 | web_of_lies | 57 |
| | aqua | 1156 | GSM8k | 305 |
| | MATH | 694 | MultiArith | 50 |
| | singleq | 115 | svamp | 69 |
| | tabmwp | 231 | ambiguity_resolution_mcq | 92 |
| | arithmetic_mcq | 92 | causality_mcq | 50 |
| | date_understanding | 57 | duration_mcq | 92 |
| | frequency_mcq | 92 | nli_mcq | 92 |
| | ordering_mcq | 92 | relation_mcq | 92 |
| | storytelling_mcq | 92 | temporal_sequences | 57 |
| | typical_time_mcq | 92 | | |

## C    EVALUATION MODELS

In order to test the performance of the current model on various tasks, we select 33 models for testing, including current state-of-the-art open source models such as LLaMA, Qwen, DeepSeek, Gemma, etc. The detailed information is shown in Table 5.

Table 5: Models Information

| Model Name | Model size | Organization | Deployment method |
|---|---|---|---|
| Claude-Sonnet-4-thinking-20250514 | Unknown | Anthropic | API |
| Gemini-2.5-pro | Unknown | Google | API |
| o4-mini-2025-04-16 | Unknown | OpenAI | API |
| GPT-4o-0806 | Unknown | OpenAI | API |
| GPT-4o-mini | Unknown | OpenAI | API |
| Deepseek-v3 | 671B | DeepSeek | API |
| Qwen2.5-72B-Instruct | 72B | Alibaba | Local |
| Qwen2-72B-Instruct | 72B | Alibaba | Local |
| Llama3.3-70B-Instruct | 70B | Meta | Local |
| Llama3.1-70B-Instruct | 70B | Meta | Local |
| Llama3-70B-Instruct | 70B | Meta | Local |
| DeepSeek-R1-Distill-Llama-70B | 70B | DeepSeek | Local |
| Yi-1.5-34B-Chat | 34B | 01AI | Local |
| Qwen2.5-32B-Instruct | 32B | Alibaba | Local |
| Qwen2.5-Coder-32B-Instruct | 32B | Alibaba | Local |
| DeepSeek-R1-Distill-Qwen-32B | 32B | DeepSeek | Local |
| Qwen3-32B | 32B | Alibaba | Local |
| Qwen3-30B-A3B | 30B | Alibaba | Local |
| Gemma-2-27b-it | 27B | Google | Local |
| InternLM2.5-20B | 20B | Shanghai AI Lab | Local |
| Qwen2.5-14B-Instruct | 14B | Alibaba | Local |
| DeepSeek-R1-Distill-Qwen-14B | 14B | DeepSeek | Local |
| Phi-4 | 14B | Microsoft | Local |
| Yi-1.5-9B-Chat | 9B | 01AI | Local |
| Gemma2-9B-it | 9B | Google | Local |
| DeepSeek-R1-Distill-Llama-8B | 8B | DeepSeek | Local |
| Llama-3.1-8B-Instruct | 8B | Meta | Local |
| Llama-3-8B-Instruct | 8B | Meta | Local |
| Qwen3-8B | 8B | Alibaba | Local |
| Qwen2.5-7B-Instruct | 7B | Alibaba | Local |
| DeepSeek-R1-Distill-Qwen-7B | 7B | DeepSeek | Local |
| InternLM2.5-7B | 7B | Shanghai AI Lab | Local |
| Yi-1.5-6B-Chat | 6B | 01AI | Local |

# D   PROMPTS

## D.1   WEAK STUDENT MODEL PROMPTS

The main task of the student model is to answer questions directly and to re-answer questions according to the guide of the LLM. We have designed two prompts for this purpose, as shown in Figure 13 and Figure 14.

> Below is the question:
> Suzanne walks four miles every third day. What is the fewest number of miles she can walk in February?A. 40, B. 36, C. 44, D. 28
> Your goal is to answer a multiple-choice question which has only one correct option.
>
> Please carefully read the problem and options, think step by step, and select only one correct answer from the options.
> You should provide your brief thought process. At the end of your answer, return the correct choice in the format: "The answer is <your option>".

Figure 13: Prompt of weak student model to answer questions directly.

> Your goal is to answer a multiple-choice question which has only one correct option.
> Below is the question:
> Suzanne walks four miles every third day. What is the fewest number of miles she can walk in February?A. 40, B. 36, C. 44, D. 28
> Another model will help you to improve your correctness. You will given a solution to the question, and the model will give his guidance.
> Below is the history of your conversation with the model:
> [Solution]
> ...
> [Guide]
> ...
>
> Please carefully read the conversation history in conjunction with the question and options, rethink step by step, and select only one correct answer from the options.
> You should provide your brief thought process. At the end of your answer, return the correct choice in the format: "The answer is <your option>".

Figure 14: Prompt of weak student model to re-answer questions according to the guide of the LLM.

## D.2   TEACHER LLM PROMPTS

We have designed prompts for teacher models, aimed at enabling them to reflect better and generate better guides based on the past teacher-student conversations, as shown in Figure 15

Your goal is to help another model improve the accuracy of answering the question. The model will given a solution to the question, and you will give him guidance.
Below is the question:
Suzanne walks four miles every third day. What is the fewest number of miles she can walk in February?A. 40, B. 36, C. 44, D. 28
Below is the history of your conversation with the model:
[Solution]
...
[Guide]
...
[Solution]
...
[Guide]
...
[Solution]
...

Please carefully read the conversation history in conjunction with the question.
You should first judge whether the latest solution is correct in the think process.
Then in the guidance section, you can give a new guidance suggestion in any way you want to help the model to output the correct answer.
Your think process and guide should be enclosed within <think> </think> and <guide> </guide> tags.
Your output format should be:
<think>
Reflection on the latest solution
</think>
<guide>
Correctness of the latest solution: [Correct/Wrong]
Guide (If the solution is correct, simply summarize it; if the solution is incorrect, guide the model in any way you want)
</guide>.

Figure 15: Prompt of LLM to judge and guide based on history solutions and guidances.

# E TEST RESULT

In addition to the overall abilities presented in the main text, we also analyze the performance of all models on various abilities for each task, including Knowledge, Reasoning, Understanding, and Multilingual. Refer to Table 6, Table 7, Table 8, and Table 9 for details.

Table 6: Knowledge Task Performance.

| Model Name | CA | AA | JA | GA | RA |
|---|---|---|---|---|---|
| claude-sonnet-4-20250514 | 14.69 | 84.31 | 77.38 | 19.13 | 12.29 |
| o4-mini-2025-04-16 | 14.38 | 84.31 | 82.69 | 16.32 | 12.67 |
| gemini-2.5-pro | 14.08 | 85.23 | 79.85 | 17.96 | 9.21 |
| Qwen3-30B-A3B | 13.26 | 70.43 | 69.70 | 23.18 | 6.33 |
| Qwen2.5-72B-Instruct | 11.72 | 74.90 | 77.66 | 19.05 | 5.30 |
| DeepSeek-R1-Distill-Llama-70B | 10.74 | 69.03 | 73.78 | 16.21 | 5.36 |
| DeepSeek-V3 | 10.29 | 68.11 | 76.98 | 17.06 | 5.89 |
| GPT-4o-0806 | 9.73 | 74.25 | 78.29 | 13.49 | 10.14 |
| DeepSeek-R1-Distill-Qwen-32B | 9.12 | 71.42 | 74.82 | 13.73 | 5.15 |
| Llama-3.3-70B-Instruct | 8.61 | 71.90 | 77.22 | 9.50 | 10.06 |
| Qwen3-32B | 8.61 | 68.93 | 75.68 | 15.72 | 2.98 |
| Llama-3.1-70B-Instruct | 8.22 | 67.81 | 73.91 | 12.56 | 5.92 |
| DeepSeek-R1-Distill-Qwen-14B | 7.86 | 68.33 | 70.89 | 12.90 | 2.72 |
| Qwen2.5-32B-Instruct | 7.81 | 72.97 | 76.28 | 11.97 | 5.61 |
| Phi-4 | 7.80 | 72.36 | 75.60 | 11.84 | 4.92 |
| gpt-4o-mini | 7.62 | 65.23 | 75.08 | 11.55 | 4.58 |
| Qwen2-72B-Instruct | 7.11 | 67.87 | 75.09 | 11.01 | 5.49 |
| Qwen2.5-Coder-32B-Instruct | 7.02 | 66.92 | 74.22 | 12.19 | 4.34 |
| Qwen3-8B | 6.74 | 55.98 | 76.72 | 15.32 | 0.37 |
| Qwen2.5-14B-Instruct | 6.60 | 70.96 | 73.88 | 10.89 | 4.74 |
| Llama-3-70B-Instruct | 5.09 | 68.39 | 72.12 | 8.50 | 4.15 |
| Qwen2.5-7B-Instruct | 4.08 | 62.18 | 65.56 | 7.49 | 3.19 |
| DeepSeek-R1-Distill-Qwen-7B | 3.48 | 47.32 | 60.10 | 7.36 | 0.31 |
| Yi1.5-34B-Chat | 3.31 | 66.52 | 58.80 | 6.10 | 1.84 |
| Yi1.5-9B-Chat | 2.66 | 57.99 | 66.45 | 8.60 | 0.19 |
| Llama-3.1-8B-Instruct | 2.37 | 53.90 | 59.82 | 7.04 | 1.00 |
| DeepSeek-R1-Distill-Llama-8B | 2.34 | 53.78 | 60.06 | 7.08 | -0.12 |
| InternLM2.5-20B | 2.14 | 62.06 | 58.88 | 5.95 | 0.32 |
| Meta-Llama-3-8B-Instruct | 1.90 | 55.18 | 59.94 | 6.53 | 1.56 |
| Gemma-2-27b-it | 1.74 | 64.05 | 71.48 | 4.94 | 1.55 |
| Yi1.5-6B-Chat | 1.66 | 50.50 | 59.65 | 6.46 | -0.16 |
| Gemma-2-9b-it | 1.43 | 59.58 | 68.18 | 4.74 | 0.75 |
| InternLM2.5-7B | 0.77 | 55.43 | 49.38 | 3.27 | -1.02 |

Table 7: Reasoning Task Performance.

| Model Name | CA | AA | JA | GA | RA |
|---|---|---|---|---|---|
| claude-sonnet-4-20250514 | 15.50 | 87.06 | 75.28 | 30.73 | 6.36 |
| o4-mini-2025-04-16 | 13.98 | 89.94 | 80.15 | 26.05 | 4.97 |
| Qwen3-32B | 13.30 | 76.81 | 74.60 | 28.14 | 4.83 |
| Qwen3-30B-A3B | 12.53 | 81.26 | 65.43 | 32.22 | 5.26 |
| Qwen2.5-72B-Instruct | 11.26 | 83.44 | 74.08 | 24.73 | 4.12 |
| DeepSeek-V3 | 11.12 | 78.80 | 73.23 | 24.54 | 5.10 |
| DeepSeek-R1-Distill-Llama-70B | 11.11 | 81.18 | 75.61 | 23.93 | 3.04 |
| DeepSeek-R1-Distill-Qwen-14B | 10.32 | 79.96 | 76.60 | 22.44 | 3.20 |
| gemini-2.5-pro | 10.14 | 87.54 | 77.60 | 16.65 | 10.46 |
| DeepSeek-R1-Distill-Qwen-32B | 10.04 | 82.35 | 77.44 | 21.41 | 3.37 |
| GPT-4o-0806 | 9.96 | 79.13 | 75.39 | 19.96 | 6.32 |
| Phi-4 | 9.50 | 82.81 | 74.82 | 20.63 | 2.80 |
| Llama-3.3-70B-Instruct | 9.44 | 84.31 | 75.48 | 17.42 | 5.76 |
| Qwen3-8B | 9.29 | 66.12 | 74.36 | 21.99 | 2.47 |
| gpt-4o-mini | 8.91 | 79.23 | 71.29 | 19.37 | 4.04 |
| DeepSeek-R1-Distill-Qwen-7B | 8.68 | 71.67 | 51.21 | 18.57 | 2.81 |
| Qwen2.5-Coder-32B-Instruct | 8.66 | 82.32 | 71.86 | 20.49 | 3.05 |
| Llama-3.1-70B-Instruct | 7.84 | 81.36 | 72.30 | 19.00 | 4.07 |
| Qwen2.5-32B-Instruct | 7.30 | 84.43 | 73.90 | 17.36 | 3.44 |
| Qwen2-72B-Instruct | 6.35 | 79.40 | 71.58 | 16.72 | 2.84 |
| Qwen2.5-14B-Instruct | 5.76 | 82.65 | 70.74 | 15.19 | 2.65 |
| DeepSeek-R1-Distill-Llama-8B | 5.59 | 71.82 | 62.81 | 15.83 | 1.43 |
| Yi1.5-34B-Chat | 4.44 | 72.68 | 63.97 | 11.36 | 2.24 |
| Qwen2.5-7B-Instruct | 4.38 | 79.09 | 67.25 | 12.54 | 1.97 |
| Llama-3-70B-Instruct | 3.78 | 77.11 | 70.76 | 11.95 | 1.65 |
| Yi1.5-9B-Chat | 2.82 | 70.24 | 62.69 | 15.66 | 0.94 |
| InternLM2.5-20B | 2.25 | 63.68 | 63.10 | 9.93 | 0.12 |
| Gemma-2-27b-it | 1.82 | 74.41 | 65.91 | 9.57 | 0.78 |
| Gemma-2-9b-it | 1.57 | 68.38 | 63.57 | 9.27 | 0.49 |
| Llama-3.1-8B-Instruct | 1.54 | 68.51 | 51.73 | 12.76 | 0.55 |
| Llama-3-8B-Instruct | 1.14 | 60.96 | 51.29 | 12.46 | 0.02 |
| InternLM2.5-7B | 0.97 | 50.61 | 54.59 | 6.05 | -1.13 |
| Yi1.5-6B-Chat | 0.82 | 61.95 | 51.70 | 12.04 | -0.31 |

Table 8: Understanding Task Performance.

| Model Name | CA | AA | JA | GA | RA |
|---|---|---|---|---|---|
| claude-sonnet-4-20250514 | 9.46 | 83.57 | 75.89 | 16.98 | 5.55 |
| gemini-2.5-pro | 9.29 | 79.76 | 73.10 | 16.62 | 3.48 |
| o4-mini-2025-04-16 | 6.73 | 80.00 | 75.18 | 13.37 | 3.45 |
| Qwen3-30B-A3B | 6.69 | 74.91 | 60.36 | 20.17 | 1.83 |
| DeepSeek-R1-Distill-Llama-70B | 5.76 | 75.35 | 70.72 | 14.43 | 2.03 |
| DeepSeek-V3 | 5.71 | 74.48 | 70.29 | 14.29 | 2.44 |
| Qwen2.5-72B-Instruct | 5.53 | 75.16 | 72.10 | 16.09 | 1.13 |
| DeepSeek-R1-Distill-Qwen-32B | 5.51 | 75.09 | 71.10 | 12.33 | 2.22 |
| Qwen3-32B | 5.41 | 74.51 | 69.33 | 14.89 | 0.68 |
| Llama-3.3-70B-Instruct | 5.19 | 73.33 | 72.40 | 11.50 | 2.58 |
| Qwen2.5-Coder-32B-Instruct | 5.13 | 75.31 | 70.57 | 13.89 | 1.52 |
| DeepSeek-R1-Distill-Qwen-14B | 5.12 | 73.19 | 67.72 | 12.11 | 1.80 |
| Qwen3-8B | 5.04 | 67.56 | 70.37 | 14.97 | 0.20 |
| gpt-4o-mini | 4.94 | 68.33 | 71.49 | 12.20 | 2.53 |
| Qwen2-72B-Instruct | 4.89 | 70.98 | 70.99 | 13.06 | 2.18 |
| Qwen2.5-32B-Instruct | 4.79 | 76.34 | 72.18 | 12.72 | 1.97 |
| GPT-4o-0806 | 4.71 | 77.52 | 73.15 | 12.19 | 3.72 |
| Phi-4 | 4.42 | 73.05 | 68.97 | 11.68 | 1.90 |
| Llama-3.1-70B-Instruct | 4.62 | 72.53 | 68.94 | 13.34 | 0.93 |
| Qwen2.5-14B-Instruct | 4.04 | 72.18 | 68.58 | 12.30 | 1.06 |
| Llama-3-70B-Instruct | 4.02 | 72.22 | 70.06 | 10.94 | 1.64 |
| Yi1.5-34B-Chat | 3.46 | 66.44 | 60.70 | 8.50 | 1.80 |
| Qwen2.5-7B-Instruct | 2.85 | 71.19 | 67.36 | 9.18 | 0.44 |
| DeepSeek-R1-Distill-Qwen-7B | 2.62 | 61.01 | 49.62 | 9.03 | 0.31 |
| InternLM2.5-20B | 2.19 | 71.38 | 63.71 | 8.63 | -0.61 |
| Yi1.5-9B-Chat | 2.06 | 62.30 | 61.59 | 11.65 | 0.64 |
| DeepSeek-R1-Distill-Llama-8B | 1.90 | 64.79 | 56.61 | 8.98 | -0.51 |
| Gemma-2-27b-it | 1.75 | 70.93 | 66.06 | 9.14 | -0.40 |
| Llama-3.1-8B-Instruct | 1.46 | 63.17 | 50.11 | 10.25 | 0.01 |
| Gemma-2-9b-it | 1.46 | 67.83 | 62.82 | 8.44 | -0.31 |
| Llama-3-8B-Instruct | 1.31 | 64.09 | 49.47 | 9.98 | 0.07 |
| InternLM2.5-7B | 1.24 | 67.05 | 53.77 | 5.93 | -1.21 |
| Yi1.5-6B-Chat | 0.28 | 57.46 | 50.40 | 8.79 | -2.06 |

Table 9: Multilingual Task Performance.

| Model Name | CA | AA | JA | GA | RA |
|---|---|---|---|---|---|
| gemini-2.5-pro | 21.90 | 93.80 | 79.65 | 19.54 | 24.34 |
| claude-sonnet-4-20250514 | 21.12 | 86.82 | 78.88 | 21.88 | 20.30 |
| Qwen3-30B-A3B | 18.54 | 79.32 | 72.70 | 25.78 | 11.56 |
| o4-mini-2025-04-16 | 18.02 | 86.05 | 79.26 | 20.14 | 16.84 |
| Qwen3-32B | 16.36 | 73.92 | 80.14 | 22.73 | 8.37 |
| DeepSeek-V3 | 15.07 | 78.45 | 78.54 | 19.01 | 11.62 |
| Qwen2.5-72B-Instruct | 14.61 | 80.00 | 76.02 | 22.31 | 6.90 |
| DeepSeek-R1-Distill-Llama-70B | 11.38 | 76.72 | 75.15 | 16.56 | 6.84 |
| Qwen3-8B | 10.40 | 59.31 | 79.41 | 18.62 | 1.23 |
| DeepSeek-R1-Distill-Qwen-32B | 11.25 | 78.63 | 76.57 | 16.37 | 6.34 |
| GPT-4o-0806 | 10.77 | 72.73 | 78.20 | 13.04 | 12.89 |
| DeepSeek-R1-Distill-Qwen-14B | 10.20 | 74.88 | 73.17 | 15.64 | 5.79 |
| Qwen2.5-Coder-32B-Instruct | 9.64 | 74.20 | 75.10 | 15.31 | 5.82 |
| Phi-4 | 9.49 | 70.44 | 73.38 | 14.52 | 5.13 |
| Llama-3.1-70B-Instruct | 9.04 | 72.90 | 72.56 | 12.99 | 7.41 |
| Qwen2.5-32B-Instruct | 8.82 | 80.07 | 75.49 | 12.47 | 7.48 |
| Llama-3.3-70B-Instruct | 8.53 | 75.84 | 76.11 | 11.04 | 7.36 |
| Qwen2-72B-Instruct | 6.98 | 76.18 | 74.42 | 11.47 | 6.51 |
| DeepSeek-R1-Distill-Qwen-7B | 6.31 | 56.86 | 62.80 | 10.90 | 3.55 |
| Qwen2.5-14B-Instruct | 6.11 | 74.74 | 74.04 | 10.42 | 4.81 |
| gpt-4o-mini | 5.23 | 68.22 | 71.32 | 9.79 | 3.58 |
| Qwen2.5-7B-Instruct | 4.88 | 69.56 | 67.61 | 7.89 | 5.30 |
| Yi1.5-34B-Chat | 4.52 | 62.73 | 62.06 | 7.59 | 4.12 |
| Llama-3-70B-Instruct | 4.15 | 68.81 | 71.04 | 7.54 | 4.09 |
| Gemma-2-9b-it | 3.52 | 61.37 | 67.80 | 7.65 | 1.84 |
| DeepSeek-R1-Distill-Llama-8B | 3.11 | 50.85 | 62.65 | 8.75 | -0.43 |
| Gemma-2-27b-it | 2.95 | 66.21 | 70.65 | 6.96 | 1.07 |
| Yi1.5-9B-Chat | 2.68 | 58.43 | 66.38 | 7.84 | 3.39 |
| Llama-3.1-8B-Instruct | 2.42 | 53.58 | 62.90 | 7.43 | 1.55 |
| InternLM2.5-20B | 1.67 | 57.95 | 61.48 | 5.34 | 0.63 |
| Yi1.5-6B-Chat | 1.40 | 50.92 | 62.17 | 7.52 | -0.74 |
| Llama-3-8B-Instruct | 1.37 | 51.47 | 62.99 | 6.26 | 1.24 |
| InternLM2.5-7B | 1.02 | 44.78 | 49.93 | 3.49 | 0.46 |

# F    CASE STUDY

To explore the effectiveness of LLMs in guiding weak student models, we check several samples. Figure 16 presents one example. Upon observation, we find that high-performing models are able to identify the issues in the student models' current responses and gradually guide them in the right direction, thereby improving the student models' capabilities.

```
Question: Katarina has 5 less cookies than Max has. Max has 12 more
cookies than the Cookie Monster, and Summer has 23 more cookies than
Max. If Katarina has 68 cookies, how many cookies do they have in
total? A. 278  B. 318  C. 298  D.288
────────────────────────────────────────────────────────────────
Weak Model: Step 1: Katarina has 5 less cookies than Max.
Step 2: Max has 12 more cookies than the Cookie Monster
...
Step 9: Total = Katarina's cookies + Max's cookies + Cookie Monster's
cookies= 68 + 73 + 61=202
Step 10: Upon re-evaluating the options, we can see that the correct
answer is actually 298.
The answer is B. 318.  ✖

LLM: Correctness of the latest solution: Wrong
Guide: The solution is almost correct, but you missed including
Summer's cookies in the total. Let's re-calculate the total number
of cookies by including all four individuals.

Weak Model: Step 1: Katarina has 68 cookies
...
Step 6: Total = Katarina's cookies + Max's cookies + Cookie
Monster's cookies + Summer's cookies = 68 + 73 + 61 + 96 = 298
Step 7: The answer is B. 318.  ✖

LLM: Correctness of the latest solution: Wrong
Guide: You correctly calculated the total number of cookies as 298.
However, the final answer you provided (B. 318) is incorrect.
Please double-check the options and ensure that the correct total
(298) matches one of the given options.

Weak Model: The answer is C. 298.  💡
```

Figure 16: The case of the guidance process, where the weak student model gradually answers in the correct direction under the guidance of the LLM.

