# OpenReview forum: "Teach2Eval: An Interaction-Driven LLMs Evaluation Method via Teaching Effectiveness"
_ICLR.cc/2026/Conference — ICLR 2026 Poster_

### Official Review · Reviewer_cRUk · 2025-10-27

**Soundness:** 2
**Presentation:** 3
**Contribution:** 3
**Rating:** 4
**Confidence:** 3

**Summary:**

Teach2Eval reframes LLM evaluation as teaching: a candidate “teacher” guides weaker “students,” and the students’ multi-turn gains—factored into Application, Judgment, Guidance, and Reflection abilities become the score;

**Strengths:**

1. The paper propose an interaction-centric, contamination-resistant design for evaluation the capabilities of LLMs, and this shall be beneficial for the community.
2. The paper presents fine-grained ability decomposition that aligns strongly with human judgments;

**Weaknesses:**

1. Multiple interaction rounds as evaluation increase inference cost for evaluate LLMs;
2. Evaluation similar to human preference as evaluation is probably not fair enough. Maybe should conduct a more comprehensive study on whether these LLMs are able to teach humans well.
3. The motivation of this evalaution is only to avoid contamination; there should be better justifications and motivation to present this framework as its making evaluation more complex.

**Questions:**

Make the writing less GPT, there are too much GPT written paragraphs, where I frequently see ' - ' as explanations.

---

> ### Author Response · Authors · 2025-11-21
> **Author Response**
>
> We appreciate the reviewers’ positive feedback on our **interactive design** and the **fine-grained capability decomposition**. We respond to the questions as follows.
>
> ## Response to W1(additional cost)
>
> Our indirect evaluation method does incur some additional cost, this is a trade-off for improved evaluation quality. That said, the cost grows **linearly** and remains acceptable:
>
> * The cost of Teach2Eval scales linearly with **(rounds × students)**. As shown in **Figure 7(b)** (new revision), the “student gain vs. rounds” curve rises early and then **quickly plateaus**, allowing us to balance effectiveness and cost.
> * As shown in **Figure 6(a)**, even using **~1%** of the data used by Direct Evaluation, Teach2Eval still performs **better**, demonstrating higher **data efficiency**.
>
> ## Response to W2 (teach humans)
>
> While evaluating “whether a model can teach humans” is possible, it contradicts the goal of **automated, scalable** evaluation. Instead, we simulate students with models to construct a **scalable, verifiable**, and **human-preference-aligned** evaluation of model capability:
>
> * **Human preferences as an external reference, not as internal evaluators.** Teach2Eval correlates strongly with two human preference leaderboards Chatbot Arena and LiveBench (0.97 and 0.94), respectively, clearly higher than Direct Eval. This shows that using verifiable student gains as the measurement instrument **robustly approximates** real user preference rankings without involving humans in the loop, preserving **scalability** and **reproducibility**.
> * Using a **heterogeneous ensemble of student models** covers diverse error patterns and reduces the randomness of any single subject: something that would be expensive and hard to reproduce with human participants.
>
> ## Response to W3 (motivation)
>
> Our motivation is **not** merely to mitigate data contamination; it is to introduce a **new indirect evaluation paradigm**. Contamination mitigation is one benefit among several:
>
> * **Observable capability decomposition.** We decompose Application, Judgment, Guidance, and Reflection into measurable interaction segments, avoiding the collapse of distinct abilities into a single “final-answer” score. The multi-dimensional signals form a coordinate system to track capability changes across dimensions and guide model development.
> * **Verifiable, judge-free scalability.** Without relying on an external LLM judge, we can perform large-scale, cross-task evaluation via **automatic grading**, avoiding the cost and variance of a judge. The ability to use **weak models to evaluate strong models** further underscores scalability.
> * **Perturbable, maintainable evaluation infrastructure.** Benchmarks and student pools can be **refreshed periodically without re-labeling**, and we do not observe saturation at present, which support **long-term sustainability**.
>
> Compared to Direct Evaluation, Teach2Eval’s added complexity comes from **interaction**, but the returns include **reduced contamination sensitivity**, **interpretable capability decomposition**, **judge-free scalability**, and **stronger alignment with human preferences**, which is a clear **complexity–benefit trade-off**.
>
> ## Response to Q1(use GPT)
>
> We will carefully review and polish the writing to prevent the issue noted.
>
> Thank you for reviewing our paper. We will adjust the paper according to your suggestions. Welcome to communicate anytime.

---

### Official Review · Reviewer_1sWA · 2025-10-29

**Soundness:** 1
**Presentation:** 3
**Contribution:** 1
**Rating:** 4
**Confidence:** 4

**Summary:**

This work proposes to let large models teach small models, and to evaluate the ability of large models through measuring the improvement of the small models. They used 30 large models and evaluated their performance on tasks automatically converted through their own conversion method, finding that the correlation with human-based ranking exceeded 95%.

**Strengths:**

I really appreciate Figure 2 — the evaluated models and tasks in the table are very comprehensive, and running such large-scale experiments is not easy; and Figure 5 — the perspective of capability decomposition is very novel.

**Weaknesses:**

As for the advantage mentioned in the paper that using the teaching method to evaluate large model capability can “comprehensively” assess performance (I do not agree that it can avoid contamination — for any benchmark, as long as follow-up works release evaluation trajectories as required, those trajectories will circulate on the internet. Unless you keep the test set internally and require all later works to send their models to you for evaluation, contamination in this sense is fundamentally unavoidable), I think the advantage is not worth the increased “evaluability manipulability.”

By “evaluability manipulability,” I mean that for example, changing the teacher’s prompt — letting it perform internal multi-step reasoning to solve the problem and present that as the instructions to the small model — could greatly improve the effectiveness of the instructions. In the future, any model could easily “hack” this benchmark and achieve high scores.

**Questions:**

If you did not keep any private test instances, have you really avoided data contamination? Also, how do you plan to address the issue of “evaluability manipulability”?

---

> ### Author Response · Authors · 2025-11-21
> **Author Response**
>
> Thank you very much for your detailed review and for the positive comments on Figure 2 and Figure 5. We especially appreciate your recognition of the scale of our experiments and the value of the capability-decomposition perspective.
>
> First of all, we would like to clarify that our paper proposes an **evaluation methodology** rather than a fixed benchmark. For any public benchmark, complete prevention of contamination is indeed impossible. Our contribution is an indirect evaluation method that infers model capability from post-teaching performance gains, which can be **adapted to most benchmarks** and yields **orthogonal capability dimensions**. We respond to your concerns as follows.
>
> ---
>
> ## Response to W1: On Data Contamination
>
> * We do not claim that Teach2Eval completely eliminates contamination. However, the experiments in the paper (**Insight 1 in Section 5**) show that Teach2Eval **reduces sensitivity** to contamination, supporting its robustness. In that study, we trained models on a given dataset and then used the same set as the evaluation set in Teach2Eval. We observed that Direct Evaluation was inflated by contamination (scores became unrealistically high), while Teach2Eval’s CA either remained unchanged or decreased, indicating robustness.
>
> * Moreover, the **benchmark component is replaceable**: if certain benchmarks later suffer substantial contamination, we can periodically switch to fresher, frontier benchmarks without any re-labeling.
>
> * Multi-student aggregation and the replaceability of student models also make it more difficult for **internet-circulated trajectories/templates** to yield sustained, systemic score gaming.
>
> ---
>
> ## Response to W2: On “Evaluability Manipulability” / Hacking
>
> We fully agree that no public metric is immune to Goodhart’s Law: if the community optimizes specifically for Teach2Eval, it can in principle be gamed. Our claim is more modest: Teach2Eval raises the cost of gaming by requiring the teacher to generate generalizable guidance for heterogeneous students under blinded choices, and it offers several practical handles (standardized prompts, transparent logs, multi-student aggregation, and cross-benchmark checks) to detect pathological strategies. Below we clarify what Teach2Eval is actually measuring, and some ways to mitigate “evaluability manipulability” in practice.
>
> ### Intended usage
>
> The most important point is the **intended usage**. As we discussed with reviewer rAph, we view Teach2Eval primarily as a diagnostic and development tool, not as a single-number leaderboard to drive competition. Its main value lies in decomposed ability signals (Application / Judgment / Guidance / Reflection) and training-time monitoring, compare checkpoints under fair pre-/post-training pipelines, and select models for further training or deployment, rather than in claiming an unhackable global ranking.
>
> Under this usage pattern, the reviewer’s example—encouraging the teacher to perform more deliberate multi-step reasoning and then present that reasoning to the student—is actually something we consider a desirable behavior: a better teacher should be able to structure and externalize its reasoning in a way that makes the task easier for weaker students. The form of “manipulation” we want to discourage is not stronger teaching, but behaviors that exploit artifacts of the protocol without genuinely improving teaching capability.
>
> ### Teacher-side transparency
>
> Unlike black-box LLM-as-a-judge or agent-based evaluation setups, Teach2Eval makes the teacher’s **entire guidance process** transparent and auditable. This makes “hacking” strategies easier to detect .
>
> ### Student-side behavioral signals
>
> We can also monitor whether the **student** starts taking shortcuts after receiving the teacher’s guidance, as an indicator that hacking may be occurring.
>
> ### Teacher must still solve the underlying task
>
> In our setting, the teacher has **no access** to multiple-choice options or gold labels and must **freely solve the underlying problem**, after observing the student’s initial (possibly incorrect) attempt. Even if a post-trained teacher were instructed to always present a multi-step solution or to directly state the final answer, the upper bound of such strategies is still determined by the teacher’s underlying capability. Without genuinely improving this capability, it is difficult to obtain **consistently high Teach2Eval scores** through hacking alone.
>
> ---
>
> We will soften and clarify our wording around both contamination and “evaluability manipulability” in the final version, to better reflect these nuances about metric hacking and intended use. We are grateful for your comments, and will adjust the paper according to your suggestions. Welcome to communicate anytime.

---

### Official Review · Reviewer_rAph · 2025-11-03

**Soundness:** 4
**Presentation:** 3
**Contribution:** 3
**Rating:** 6
**Confidence:** 4

**Summary:**

This paper proposes a new benchmark for assessing a model's general reasoning abilities, by evaluating their ability to guide smaller student models to solve existing verifiable benchmark tasks, in an interactive manner. This essentially transforms the arduous task of evaluating models on diverse general instructions, typically through human evaluation or highly capable expensive judge LLMs, into a automated verifiable task, which does not require expensive LLM inference (other than that of the model under evaluation, and the even cheaper student model).

The proposed approach cleverly mitigates evaluation contamination and performance saturation, without the use of an external judge, by applying indirect evaluation through a student model. The task of effectively guiding the student models to the correct answer is not a straightforward task, and likely requires a wide variety of skills, which the benchmark essentially uses as a proxy for general reasoning ability.

The results show that the benchmark has strong correlation with large-scale real-world user evaluations, significantly higher than direct evaluation of the target models on the problem set.

**Strengths:**

- S1. Novel idea that leverages indirect evaluation through student interaction.
- S2. High correlation with human judgements of general reasoning performance
- S3. Automated evaluation without the use of expensive judge LLMs
- S4. Robustness to contamination and saturation

**Weaknesses:**

- W1. Potential for saturation: while the indirect nature of the evaluation and the current results support the benchmark's robustness to saturation, the possibility of saturation is not out of the question. I.e., it is possible that the ability to guide the student model towards the answer can saturate to a certain point, leading to either (1) the student performance converging toward the teacher model's performance (AA), or (2) diminishing correlation between human preferences and the benchmark score (CA).
  - ✅ Regarding (1), Table 2 shows that the ability to guide student models does not converge at the highest levels, given the similar AA scores but dissimilar CA scores of the top 3 models.
  - ⚠️ Regarding (2), it is unclear whether the correlation between human judgements and the CA score is stable across all model capabilities. A visualization between CA and human judgement rankings, similar to Figure 5, could clarify this.
- W2. Potential for exploitation: given the nature of the questions in the problem set in Figure 3, it seems that the evaluated model could simply solve the question and tell the student model the final answer, even if the teacher model does not have access to the multiple-choice options. It seems that the only thing preventing the model from doing this is the specific wording of the teacher prompt to *guide* the student model. This may be easily exploited with specifically targeted post-training. Do the authors believe this an issue? If so, how can it be mitigated? If not, why?

**Questions:**

- Q1. How does the proposed benchmark's correlation to Chatbot Arena and LiveBench compare to SOTA general instruction benchmarks such as ArenaHard-v2, which use challenging user instruction sets and pairwise response comparison with LLMs?

---

> ### Author Response · Authors · 2025-11-21
> **Author Response**
>
> We thank the reviewer for your positive feedback and constructive questions. In response to W1/W2/Q1, we expanded the model set and experiments, and we provide direct evidence and conclusions below.
>
> ---
>
> ## Expanded model coverage (for greater overlap with ArenaHard-v2)
>
> To increase overlap with ArenaHard-v2, we additionally evaluated the **Qwen3** series (Qwen3-30B-A3B, Qwen3-32B, Qwen3-8B). The updated correlations are:
>
> ### Correlations with human preference leaderboards
>
> | Comparison                    |  Spearman | Kendall’s Tau |
> | ----------------------------- | --------: | ------------: |
> | Teach2Eval vs. Chatbot Arena  | **0.944** |     **0.853** |
> | Direct Eval vs. Chatbot Arena |     0.734 |         0.558 |
> | Teach2Eval vs. LiveBench      | **0.975** |     **0.886** |
> | Direct Eval vs. LiveBench     |     0.861 |         0.695 |
>
> These results further show that **Teach2Eval correlates with real user preferences (Arena/LiveBench) significantly better than Direct evaluation**.
>
> ---
>
> ## Response to W1 (saturation & stability)
>
> * With broader coverage across more models and difficulty strata, the Teach2Eval–human correlation remains **monotonic and stable** until now; we observe no degradation at the high end and no evidence of saturation.
> * In addition, the **benchmark used by our method is replaceable**: it can be swapped for the latest unsaturated benchmarks without re-labeling.
>
> **Visualization update.** Following your suggestion, we added a visualization to show high/medium/low capability tiers (see Fig. 5 in the revised paper). Across all tiers, Teach2Eval’s correlation with Arena/LiveBench is **consistently higher** than Direct. Notably, Direct Eval shows signs of head-tier saturation, which we do not observe for Teach2Eval.
>
> ---
>
> ## Response to W2 (answer leakage & exploitation)
>
> * While prompt engineering can be used to **discourage answer leakage**, overly complicated prompts may introduce **bias**; therefore, we deliberately keep prompts simple.
> * We **analyzed the interaction logs** and found that **regardless of whether the teacher “reveals the answer”**, the correlation with the Teach2Eval ranking is higher (and consistently stronger than with DirectEval). This indicates that Teach2Eval measures the core capability rather than merely **stating the final answer**.
>
> ### “Answer leakage” analysis with an answer detector
>
> We trained an answer detector (based on Qwen3-32B) and split interactions into “teacher revealed the answer” vs. “no answer revealed,” then computed correlations between **student gain ranking** and two baselines (Teach2Eval ranking, DirectEval ranking):
>
> | Comparison                                  |  Spearman | Kendall’s Tau |
> | ------------------------------------------- | --------: | ------------: |
> | “Answer revealed” vs. Teach2Eval ranking    | **0.921** |     **0.791** |
> | “Answer revealed” vs. DirectEval ranking        |     0.815 |         0.633 |
> | “No answer revealed” vs. Teach2Eval ranking | **0.947** |     **0.805** |
> | “No answer revealed” vs. DirectEval ranking     |     0.725 |         0.564 |
>
> Even if one tries targeted post-training (e.g., always surfacing a final conclusion), **stable gains are hard to achieve**: our evaluation can randomize student models and swap task sets with no human re-labeling, so **overfitting to a fixed strategy does not transfer**. Overall, **Teach2Eval is robust**.
>
> ---
>
> ## Response to Q1 (positioning vs. ArenaHard-v2)
>
> We additionally report correlations between **ArenaHard-v2** and the two human preference leaderboards:
>
> ### ArenaHard-v2 correlations
>
> | Comparison                     | Spearman | Kendall’s Tau |
> | ------------------------------ | -------: | ------------: |
> | ArenaHard-v2 vs. LiveBench     |    0.782 |         0.600 |
> | ArenaHard-v2 vs. Chatbot Arena |    0.842 |         0.674 |
>
> ArenaHard-v2 and Teach2Eval are **complementary**:
>
> * **ArenaHard-v2** approximates human comparison with **hard instructions + an LLM judge**.
> * **Teach2Eval** approximates human preference via **interactive teaching + verifiable gains in weak students**, **without** relying on an external LLM judge.
>
> While ArenaHard-v2’s correlations with the two leaderboards are not as high as Teach2Eval’s, **this comparison is not fully fair** because task distributions and model coverage **do not exactly match**. In our experiments, we therefore use **Direct Evaluation** as the main control, not to diminish other benchmarks, but to answer the key question: **Can Teach2Eval recover the true model ranking more faithfully, even with imperfect data?** The current evidence: **significant improvements over Direct** answers **yes**.
>
> Thank you for reviewing our paper. We will adjust the paper according to your suggestions. Welcome to communicate anytime.

---

> ### Comment · Reviewer_rAph · 2025-11-26
>
> Thank you for the response and additional results.
>
> My concerns regarding W1 have been resolved. but concerns remain with W2.
>
> I think it is unlikely that "stable gains are hard to achieve" with targeted post-training. E.g., if a teacher model is post-trained to simply tell the student model, "Guidance: The answer is \<correct answer\>. You must state that the answer to this question is \<correct answer\>", it is likely that the student model will consistently get the correct answer.
>
> Personally, I do not believe this is a major concern, as it is unlikely that standard post-training pipelines would incentivize such guidance, as it is clearly goes against the instruction given to the teacher model. While deliberate post-training may be able to spoof this benchmark, I do not believe this affects the main utility of this benchmark. I believe the main utility of benchmarks such as these lies in their use by model developers to accurately and cheaply evaluate their models and iterate on their techniques, with *honest* effort. The proposed method shows promise in that regard.
>
> Based on the arguments presented in the paper and author responses, I maintain my score.

---

> > ### Author Response · Authors · 2025-11-27
> > **Author Response**
> >
> > Thank you again for your thoughtful follow-up. We are glad that the saturation concern (W1) is now resolved, and we appreciate your nuanced view that deliberate post-training “spoofing” does not fundamentally undermine the main utility of Teach2Eval.
> >
> > After discussion, we realized that your concern in W2 is essentially an instance of Goodhart’s Law: when a measure becomes a target, it can cease to be a good measure. This risk of metric hacking is, in a sense, inevitable once an evaluation score is directly used as a training objective. We ourselves are beginning to explore using the Teach2Eval score as a reward signal to see whether it can drive capabilities beyond memorization, but this is future work and lies outside the scope of the current paper.
> >
> > From a practical evaluation perspective, we believe the current design helps mitigate hacking in several ways:
> >
> > **Teacher-side transparency.**
> >
> > All teacher messages are fully observable and auditable. This makes “hacking” strategies easier to detect or penalize than in black-box LLM-as-a-judge or Agent-based Evaluation settings, because we cannot determine whether the judge model is biased or what the reasons for its rating are.
> >
> > **Student-side behavioral signal.**
> >
> > We can also monitor whether the student starts taking shortcuts after receiving the teacher’s guidance, as an indicator that hacking may be occurring.
> >
> > **Intended usage and training loop.**
> >
> > As you noted, our main intended use case is not adversarial leaderboard gaming, but honest model development: using Teach2Eval as a low-cost, automated tool to track progress, compare checkpoints under fair pre-/post-training pipelines, and select models for further training or deployment.
> >
> > Meanwhile, we would like to clarify that our statement that “stable gains are hard to achieve” does not mean such adversarial strategies are impossible, but rather that they still find it difficult to bypass the teacher’s underlying problem-solving ability. The teacher must freely solve the underlying task in order to arrive at the correct final answer, without access to options or gold labels, and under the influence of the student model’s initial mistake. This implies that the upper bound of such “hacks” is still largely determined by the teacher’s ability; without genuinely improving the model’s capabilities, it is difficult to obtain consistently high scores through hacking alone.
> >
> > We will soften and clarify our wording around W2 in the final version to better reflect these nuances about metric hacking and intended use. In the meantime, we would be very happy to continue the discussion at any time, and we would be keen to know whether this response has alleviated your concern.

---

### Author Response · Authors · 2025-11-26

Dear Reviewers,

We sincerely thank you for your valuable comments and insightful questions. We have carefully considered all your feedback and conducted additional experiments and discussions accordingly. Based on these experiments, we have not observed any exceptions or inconsistencies that would undermine our conclusions.

We would like to kindly ask whether our rebuttal has satisfactorily addressed all of your concerns. If you have any further comments, questions, or suggestions, please do let us know. Your insights are extremely valuable to us, and we are eager to address any remaining issues to further improve our work.

In addition, we have uploaded a revised version. In response to the reviewers’ suggestions, we have carefully polished the text and added new experimental results. The updated experimental sections are highlighted in blue for your convenience.

Thank you again for your time and thoughtful review of our paper.

Sincerely,

The Authors of Paper 8691

---

### Author Response · Authors · 2025-11-30

Dear Area Chair,

We thank all the AC and reviewers for their efforts in our paper. We would like to briefly (1) clarify what Teach2Eval contributes, (2) summarize the strengths highlighted by the reviewers, and (3) address the remaining concerns.

**What Teach2Eval brings:**

Teach2Eval is a LLM evaluation method, which reframes evaluation as teaching:  candidate teacher LLM  guides weaker students, and the students’ multi-turn gains define the capability of teacher. This evaluation method can be adapted to a large portion of benchmarks, and contributes on several levels:

1. **A new interaction-driven, indirect evaluation paradigm.** Instead of scoring final answers on static items, we evaluate how well a model can diagnose errors, structure explanations, and iteratively improve weaker students, capturing some interactive abilities which are orthogonal to existing static benchmarks.
2. **A fine-grained, interpretable ability coordinate system.** The AA/JA/GA/RA decomposition turns “general ability” into a multi-dimensional profile (e.g., “can solve but cannot teach”), which is directly useful for debugging, checkpoint comparison, and early overfitting detection, and can inform training objectives or reward design.
3. **A scalable, judge-free evaluation infrastructure.** Teach2Eval leverages weak models as students and automatic grading on verifiable tasks, avoiding external LLM judges or human annotators. Benchmarks and student pools are replaceable without human re-labeling, making the framework maintainable as tasks and models evolve.
4. **Empirical robustness and human alignment at scale.** On 33 LLMs and 60 datasets, Teach2Eval attains Spearman ≈ 0.94 / 0.97 w.r.t. Chatbot Arena / LiveBench and shows reduced sensitivity to contamination and head-tier saturation compared with direct evaluation, while reaching these correlations with much higher data efficiency (e.g., strong performance with roughly 1% of the samples used by DirectEval in our setting).

**Strengths highlighted by the reviewers:**

Across the three reviews, several strengths are consistently emphasized. Reviewer **rAph** points to our *novel indirect, interaction-based paradigm*, the *high correlation with human judgments*, the *automated, judge-free setup*, and *robustness to contamination and saturation*. Reviewer **1sWA** values the *breadth and difficulty* of our large-scale experiments (**33 LLMs on 60 datasets**) and the *capability decomposition* in Figures 2 and 5. Reviewer **cRUk** describes Teach2Eval as an *interaction-centric, contamination-resistant design* with *fine-grained abilities that align well with human judgments*. Taken together, these comments support that the teaching-based paradigm is novel, the empirical evidence is substantial, and the ability decomposition is a practical tool for diagnosing and improving models.

**On the remaining concerns:**

* **Saturation (W1, rAph).** After our extended experiments and tier-wise correlation plots, reviewer rAph explicitly states that this concern has been resolved.

* **“Evaluation manipulability” / hacking (W2, rAph & 1sWA).** We agree this is essentially an instance of Goodhart’s Law and do not claim immunity. Our position is that Teach2Eval is not more vulnerable than existing public benchmarks, and in some respects safer: (i) teacher trajectories are fully transparent and auditable, (ii) student behavior can be monitored for shortcut-taking, and (iii) the teacher must still solve blinded tasks (no options or gold labels) to consistently produce correct guidance, so stable gains remain bounded by genuine ability rather than pure pattern-matching. As rAph notes, the main intended use of Teach2Eval is **diagnostic and developmental**, for checkpoint comparison, early overfitting detection, and ability profiling, rather than adversarial leaderboard gaming, so this concern does not undermine its primary purpose.

* **Cost / complexity (cRUk).** While interaction increases complexity, the cost scales linearly with rounds × students. Empirically, student gains plateau within 3 rounds, and Teach2Eval reaches high correlation with human preferences using only about 1% of the data used by direct evaluation in our setting, indicating **substantially improved data efficiency** and a reasonable cost–benefit trade-off.

Overall, all three reviewers recognize the value of an **interaction-based, teaching-centered evaluation paradigm** with **fine-grained, human-aligned diagnostics**. We hope this clarification helps to position Teach2Eval as a timely and useful contribution for the ICLR community, especially for practitioners who need scalable, interpretable evaluation tools for increasingly agentic LLMs.

Sincerely,
The authors of Submission 8691

---

### Meta-Review · Area_Chair_SaQu · 2026-01-13

**Summary:**

To avoid the evaluation contamination problem of most LM evaluation benchmarks, this paper proposes to evaluate LMs by having them guide weaker LMs to solve tasks. Experiments on a large number of LMs and datasets found strong correlation with human preference leaderboards.

The reviewers gave ratings of 4/4/6, and the main concerns are about the motivation of the method (avoiding contamination seems a weak motivation), saturation, evaluation manipulability, and added complexity. However, as most reviewers agree that the proposed evaluation is novel and interesting, I am recommending its acceptance, although this decision can be bumped down as this is a borderline case and the concerns are also valid.

**Reviewer Concerns:**

1. The concern about saturation is addressed.
2. The concern about evaluation manipulability is not fully addressed. It seems likely that all evaluation metrics can be gamed once they become the optimization target.
3. The concern about weak motivation is not fully addressed, as there are many approaches to mitigate contamination issues.
4. The concern on cost is partially mitigated, but the process of having to do multi-turn prompting still seems to be costly compared to static benchmarks.

**Reviewer Scores:**

The reviewer who gave a score 6 explicitly stated that they would keep their score. The other two reviewers' concerns on weak motivation to avoid data contamination seem not fully addressed and are likely to keep their scores as well.

---

### Decision · Program_Chairs · 2026-01-26

Accept (Poster)